# On-policy Distillation of Language Models: Learning from Self-Generated Mistakes

**Rishabh Agarwal**[12*]    **Nino Vieillard**[1*]    **Yongchao Zhou**[13]

**Piotr Stanczyk**[1†]  **Sabela Ramos**[1†]  **Matthieu Geist**[1]  **Olivier Bachem**[1]

[1]**Google DeepMind**   [2]**Mila**   [3]**University of Toronto**

## Abstract

Knowledge distillation (KD) is widely used for compressing a teacher model to reduce its inference cost and memory footprint, by training a smaller student model. However, current KD methods for auto-regressive sequence models suffer from distribution mismatch between output sequences seen during training and those generated by the student during inference. To address this issue, we introduce Generalized Knowledge Distillation (GKD). Instead of solely relying on a fixed set of output sequences, GKD trains the student on its self-generated output sequences by leveraging feedback from the teacher on such sequences. Unlike supervised KD approaches, GKD also offers the flexibility to employ alternative loss functions between the student and teacher, which may be useful when the student lacks the expressivity to mimic the teacher's distribution. Furthermore, GKD facilitates the seamless integration of distillation with RL fine-tuning of language models. We demonstrate the efficacy of GKD for distilling auto-regressive T5 language models for task-specific distillation on summarization, translation, and reasoning tasks, and task-agnostic distillation for instruction tuning.

## 1 Introduction

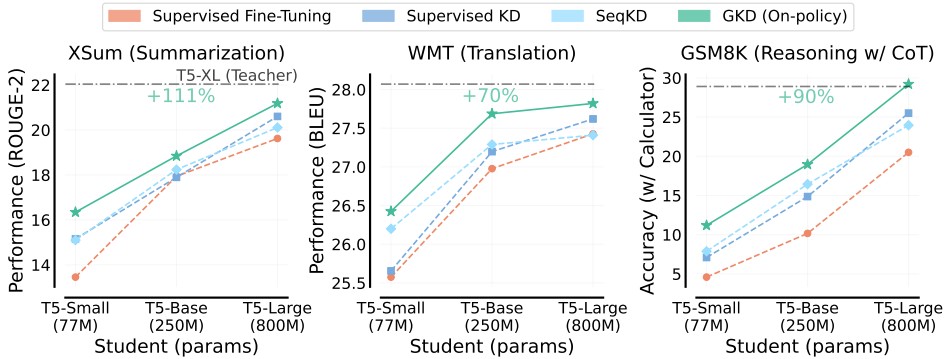

Figure 1: **Comparing GKD with KD approaches** across different student model sizes. We use the T5 models ([Raffel et al., 2020](#)) trained with supervised FT as students. We use supervised FT T5-XL (∼3B params) as the teacher, whose performance is indicated by the horizontal line. Supervised KD and FT use ground-truth output sequences for training while SeqKD trains on output sequences generated by the teacher. On-policy GKD trains on output sequences sampled from the student. For GKD, we use JSD (0.1) on WMT and forward KL on other tasks. For evaluation, we use greedy sampling for XSum and GSM8K and beam search for WMT.

Auto-regressive sequence models, such as language models (LMs), have shown impressive capabilities in numerous tasks, where the key to this success is often scaling the amount of training data as well as the number of model parameters ([Kaplan et al., 2020](#)). However, scaling parameter count comes at a

---

*denotes equal contribution. †denotes infrastructure contribution. Correspondence to Rishabh Agarwal <rishabhagarwal@google.com> and Nino Vieillard <vieillard@google.com>.

cost, and the deployment of such models is limited by either their inference cost or memory footprint. Thus, a crucial goal for practical use of large capable models is to compress them by reducing their parameter count, while retaining as much as possible of their performance.

One of the prevalent techniques for compressing models is knowledge distillation (Hinton et al., 2015). Distillation is the process of training a model – the student – to replicate the knowledge of another model – the teacher – on a specific set of tasks. Typically, the student has fewer parameters than the teacher and as such, distillation can improve task-specific performance while maintaining lower inference cost and memory footprint than the teacher. Current distillation methods for autoregressive sequence models either require generating a fixed set of output sequences from the teacher model (Kim & Rush, 2016), which can be expensive, or a fixed dataset of sequences that the teacher can label by assigning token-level probabilities (Sanh et al., 2019). However, using a fixed dataset can lead to distribution mismatch between output sequences seen during training and the sequences generated by the student auto-regressively during inference, a well-known problem in imitation learning (Pomerleau, 1991; Ross & Bagnell, 2010). Furthermore, the common objective for distillation is to minimize the forward KL between the teacher and the student distributions. However, the student may not be expressive enough to fit the teacher's distribution, which can result in student-generated samples that are unlikely to be generated by the teacher (*e.g.*, Figure A.16).

In this paper, we propose Generalized KD (GKD) to mitigate the above issues. First, we recognize that KD for auto-regressive sequence models can be viewed as an imitation learning problem with an interactive expert (Ross et al., 2011). Using this insight, GKD trains the student on its self-generated sequences that are on-policy, instead of a fixed set of outputs sequences, using teacher probabilities as expert labels on these sequences. Our idea is further supported by the recent success of fine-tuning large language models on their own output sequences (Ouyang et al., 2022; Singh et al., 2023). Furthermore, GKD provides the flexibility to optimize alternative divergence measures, such as reverse KL and generalized JSD (Section 2), that can use student's limited capacity to focus on generating samples that are likely under the teacher.

GKD unifies some existing KD methods for autoregressive LMs while instantiating new on-policy methods that substantially outperform prevalent approaches. In terms of performance gains over the initial student from on-policy GKD, averaged across T5 student models of different sizes, we see relative gains of $2.1\times$ on summarization, $1.7\times$ on machine translation, and $1.9\times$ on arithmetic reasoning tasks, compared to the performance improvements achieved with baseline KD methods (Figure 1). Additionally, we exhibit GKD's efficacy in task-agnostic distillation, resulting in 2% and 1% **absolute** accuracy improvement on the held-out BBH and MMLU benchmark suites (Figure A.11).

Our key contributions are:

- To tackle discrepancy during training and inference for auto-regressive LMs, we present GKD that leverages on-policy student-generated outputs for distillation, guided by the token-level teacher probabilities over these outputs. GKD substantially outperforms commonly-used methods in task-specific (Figure 1) and task-agnostic KD (Figure A.11).

- We demonstrate that on-policy GKD can be seamlessly combined with RL fine-tuning (e.g., RLAIF) of language models, a combination that has not been previously explored (Figure 5).

- Through a systematic evaluation of design choices in GKD, we offer practical insights about the importance of using student-generated on-policy output sequences during distillation and the task-dependent nature of optimal divergence between the student and the teacher.

## 2 PRELIMINARIES

**Auto-regressive Generative Sequence Models**. We denote the input and output sequence as $x, y$ respectively. Let $\mathbb{V}$ denote the vocabulary comprising of $M$ tokens, $y_{<n+1} = (y_1, y_2, \ldots, y_n)$ denote the generated output sequence up to the $n^{th}$ token, and $L_y$ denote the length of sequence $y$. A token-level auto-regressive policy $p(.|y_{<n}, x) \in (0, 1)^M$ outputs a next-token probability distribution over all tokens in $\mathbb{V}$, conditioned on the input $x$ and output sequence $y_{<n}$. Furthermore, $y \sim p(\cdot|x)$ corresponds to a sampled output sequence $y$ given the input $x$. For ease of notation, we define $p(y_n|x) := p(y_n|y_{<n}, x)$. Auto-regressive generation involves predicting tokens one at a time, based on the previously generated tokens. The probability of predicting $n^{th}$ token $y_n$, $p(y_n|x)$, is determined using a softmax with temperature $\gamma$: $p(y_n|x) = \frac{\exp(z_n/\gamma)}{\sum_{i=1}^{M} \exp(z_i/\gamma)}$, where $z_n$ is the logit score for the

token $y_n$. Higher values of $\gamma$ introduces more randomness, while a lower value makes the output more deterministic by favoring the most probable words. During training, the student's temperature is kept at 1. For evaluation, we use *greedy sampling* ($\gamma \to 0$) or *temperature sampling* ($\gamma > 0$).

**KL-Based Divergences**. The divergence between two probability distributions is a measure of the similarity of the distributions, with KL divergence a prevalent measure. The KL divergence between two discrete distributions $P(\mathcal{C})$ and $Q(\mathcal{C})$ is given by: $\mathcal{D}_{KL}(P\|Q) = \sum_{c\in\mathcal{C}} P(c) \log \frac{P(c)}{Q(c)}$.

The KL divergence is not symmetric: $\mathcal{D}_{KL}(P\|Q) \neq \mathcal{D}_{KL}(Q\|P)$. As such, we refer to $\mathcal{D}_{KL}(P\|Q)$ as the **forward KL** while $\mathcal{D}_{KL}(Q\|P)$ as the **reverse KL** between $P$ and $Q$. Forward KL under an empirical data distribution corresponds to maximum likelihood, which we optimize in supervised learning. Given model capacity mismatch, when approximating $P(\mathcal{C})$ using a distribution $Q_\theta(\mathcal{C})$, minimizing the reverse and forward KL results in mean and mode-seeking behavior (Figure A.16).

While KL divergence can be unbounded, a well-known divergence that is *bounded* even for probability distributions with disjoint supports is the **generalized JSD** (Jensen-Shannon divergence). JSD($\beta$) interpolates between the forward and reverse KL using the bounded coefficient $0 < \beta < 1$:

$$\mathcal{D}_{JSD(\beta)}(P\|Q) = \beta\mathcal{D}_{KL}\Big(P\Big\|\beta P + (1-\beta)Q\Big) + (1-\beta)\mathcal{D}_{KL}\Big(Q\Big\|\beta P + (1-\beta)Q\Big) \quad (1)$$

Huszár (2015) show that $\lim_{\beta\to 0} \mathcal{D}_{JSD(\beta)}(P\|Q)/\beta = \mathcal{D}_{KL}(P\|Q)$. As such, gradients of JSD($\beta$) behave similarly to forward KL and reverse KL when $\beta$ is close to 0 and 1 respectively.

## 3 DISTILLATION FOR AUTO-REGRESSIVE SEQUENCE MODELS

**Problem Setup**. We are given two auto-regressive sequence models of different capacity, where $p_S$ and $p_T$ refers to the student and teacher respectively. We assume that the student has learnable parameters $\theta$ and $p_S^\theta$ is differentiable w.r.t $\theta$. We are also given a dataset of inputs $X$. Optionally, we can also assume access to a dataset of input-output sequence pairs $(X, Y)$. If not given, such a dataset can be generated by sampling sequences from the teacher. For a divergence $\mathcal{D}$, we define the discrepancy between token-level distributions of $p_T$ and $p_S$ as

$$\mathcal{D}\big(p_T\|p_S^\theta\big)(y|x) := \frac{1}{L_y} \sum_{n=1}^{L_y} \mathcal{D}\big(p_T(\cdot|y_{<n}, x)\|p_S^\theta(\cdot|y_{<n}, x)\big), \quad (2)$$

for an input $x$ and output sequence $y$. For example, using JSD($\beta$) as $\mathcal{D}$ in equation 2 results in $\mathcal{D}_{JSD(\beta)}\big(p_T\|p_S^\theta\big)(y|x) = \frac{1}{L_y} \sum_n \mathcal{D}_{JSD(\beta)}\big(p_T(\cdot|y_{<n}, x)\|p_S^\theta(\cdot|y_{<n}, x)\big)$.

**Supervised FT**. If we are only given a fixed dataset of ground-truth output sequences but not query access to the teacher policy, then a simple approach is to minimize the negative log-likelihood of such sequences under the student policy: $L_{SFT}(\theta) = \mathbb{E}_{(x,y)\sim(X,Y)}\big[-\log p_S^\theta(y|x)\big]$.

**Sequence-Level KD** (Kim & Rush, 2016). SeqKD maximizes the likelihood of high probability sequences generated by the teacher, and can be viewed as supervised FT on teacher-generated outputs.

**Supervised KD** (Hinton et al., 2015; Sanh et al., 2019) is a widely used technique where the student is trained to imitate the token-level probability distributions of the teacher. The student $p_S$ is trained with the supervised objective $L_{SD}$ over the target token-level probabilities of the teacher $p_T$:

$$L_{SD}(\theta) := \mathbb{E}_{(x,y)\sim(X,Y)}\Big[\mathcal{D}_{KL}\big(p_T\|p_S^\theta\big)(y|x)\Big], \quad (3)$$

where the expectation is over the samples from the dataset. This supervised objective results in a rich training signal by leveraging the full token-level distribution of the teacher.

### 3.1 GENERALIZED KNOWLEDGE DISTILLATION (GKD)

As discussed above, commonly-used KD approaches use a fixed dataset of output sequences, either using ground-truth targets or teacher-generated sequences. However, distilling auto-regressive student models using such approaches results in train-inference distribution mismatch. This is because the partial sequences encountered by the student during the auto-regressive generation phase at inference can be quite different from the ones seen during the training phase. Since predictions at any step are contingent upon previous steps in auto-regressive models, this mismatch can have a cascading effect

---

**Algorithm 1** Generalized Knowledge Distillation (GKD)

1: **Given**: Teacher model $p_{\text{T}}$, Student Model $p_{\text{S}}^{\theta}$, Dataset $(X, Y)$ containing (input, output) pairs
2: **Hyperparameters**: Student data fraction $\lambda \in [0, 1]$, Divergence $\mathcal{D}$, Learning rate $\eta$
3: **for** each step $k = 1, \ldots, K$ **do**
4:     Generate a random value $u \sim Uniform(0, 1)$
5:     **if** $u \leq \lambda$ **then**
6:         Sample inputs $x$ from $X$ and generate outputs $y \sim p_{\text{S}}^{\theta}(\cdot|x)$ to obtain $B = \{(x_b, y_b)\}_{b=1}^{B}$
7:     **else**
8:         Sample batch of inputs and outputs from $(X, Y)$ to obtain $B = \{(x_b, y_b)\}_{b=1}^{B}$.
9:     **end if**
10:    Update $\theta$ to minimize $L_{\text{GKD}}$: $\theta \leftarrow \theta - \eta \frac{1}{B} \sum_{(x,y) \in B} \nabla_\theta \mathcal{D}(p_{\text{T}} \| p_{\text{S}}^{\theta})(y|x)$
11: **end for**

---

where error in prediction at early step can affect the future predictions, resulting in poor quality text generation. To address this mismatch, we draw heavily from imitation learning (IL). In particular, on-policy imitation approaches (*e.g.* Ross et al., 2011) iteratively collect sequences using the student policy, obtain expert labels for those sequences, and then retrain the student on this dataset. Despite their popularity in robotics and deep RL (Parisotto et al., 2015; Kelly et al., 2019; Agarwal et al., 2022), on-policy approaches are typically not used for distilling auto-regressive models.

Extending on-policy imitation to distillation, we present **on-policy KD**. When using on-policy data during distillation, the student receives token-specific feedback from the teacher's logits on the erroneous tokens in its self-generated output sequences. This enables a form of feedback loop akin to what we observe in RL, which helps minimize the train-inference distribution mismatch. Moreover, as the student evolves during training, the data it generates also improves in quality. Given an input $x$, the student generates the output sequence $y$ and imitates the teacher token-level distributions, $p_T(y_n|x)$, on intermediate states $y_{<n}$. Specifically, the on-policy loss $\mathcal{L}_{OD}$ is given by

$$L_{OD}(\theta) := \mathbb{E}_{x \sim X}\left[\mathbb{E}_{y \sim p_{\text{S}}(\cdot|x)}\left[\mathcal{D}_{KL}\big(p_{\text{T}} \| p_{\text{S}}^{\theta}\big)(y|x)\right]\right], \tag{4}$$

where we do *not* backpropagate through the student's sampling distribution $p_{\text{S}}(\cdot|x)$, similar to on-policy imitation. Not backpropagating through the sampling makes the training stable and computationally efficient. In on-policy KD, the training is done on output sequences that the student is likely to generate. During training, we use a temperature of $\gamma = 1$ to encourage diversity in student generated sequences. Moreover, given unlabeled input prompts, generating sequences using the student is computationally cheaper than the teacher, due to differences in their model sizes.

Building further upon on-policy KD, we unify supervised and on-policy approaches and propose a more general approach, which we call Generalized KD (**GKD**). In GKD, we can choose both the divergence to optimize as well as the output sequences to train on. Specifically, we can optimize any divergence between the teacher and student token-level probability distributions. For output sequences, GKD uses a mixture of fixed dataset, either teacher-generated or ground-truth, and on-policy student-generated sequences. Abstractly, GKD minimizes an objective of the form:

$$L_{\text{GKD}}(\theta) := (1-\lambda)\mathbb{E}_{(x,y)\sim(X,Y)}\big[\mathcal{D}(p_{\text{T}}\|p_{\text{S}}^{\theta})(y|x)\big] + \lambda \mathbb{E}_{x\sim X}\left[\mathbb{E}_{y\sim p_{\text{S}}(\cdot|x)}\big[\mathcal{D}(p_{\text{T}}\|p_{\text{S}}^{\theta})(y|x)\big]\right],$$

where $D(p_{\text{T}}, p_{\text{S}})(y|x)$ is a divergence between teacher and student distributions (equation 2), and $\lambda \in [0, 1]$ is a hyper-parameter that controls the *student data fraction*, that is, the fraction of on-policy student-generated outputs. Akin to on-policy KD, we do not backpropagate gradients through the student's sampling process. On-policy and supervised KD are instantiations of GKD with divergence $\mathcal{D}$ set to forward KL and student data fractions $\lambda$ to 1 and 0 respectively. That said, GKD allows for other choices for the fraction $\lambda$ and the divergence, which we explore in this work.

**Remark**. As opposed to a randomly initialized student, we assume access to a student that can generate sequences of adequate quality, which the teacher can provide feedback upon. In our experiments, we start from student models that have undergone supervised FT. This is analogous to two-stage RLHF training, which is widely used for LMs, where we first run SFT followed by the online RL fine-tuning. As such, GKD can leverage hyperparameter tuning insights from RLHF and can be combined with RLHF with small compute overhead and no additional hyperparameters.

*Choice of Divergence in GKD*. While forward KL is commonly-used for distillation, it requires the student to cover the entire support of the teacher token-level distribution $p_{\mathrm{T}}(.|y_{<n}, x)$. In doing so, the student might end up assigning probability mass to tokens $v$ which have low probability under $p_{\mathrm{T}}(.|y_{<n}, x)$, which can result in hallucination and low-quality generations. When the student has much lower model capacity than the teacher, this issue is likely to happen with temperature sampling (*e.g.*, Figure A.16). Alternatively, mode-seeking divergences, such as reverse KL, prioritize the tokens where the teacher assigns high probability, which can avoid low-quality generations but at the expense of less diverse generations for a given input. Our experiments indicate that optimal divergence seems to be task-dependent. Overall, the diversity and performance trade-offs for a particular task needs to be considered when choosing the GKD divergence (*e.g.*, Figure 4, A.11).

## 3.2 RL FINE-TUNING + ON-POLICY GKD

In some tasks, it is plausible that distilling from a teacher model only provides a proxy to our main objective, which can also be non-differentiable. We can directly optimize this objective with reinforcement learning (RL). Conveniently, on-policy GKD can be easily combined with RL fine-tuning from human (RLHF) or AI feedback (RLAIF), as it only requires output samples from the student. Indeed, consider that one wants to optimize the student policy for a scalar reward $r$, while staying close to a teacher policy, then we get a regularized RL fine-tuning objective of the form:

$$\mathbb{E}_{x \sim X}\Big[(1-\alpha)\underbrace{E_{y \sim p_{\mathrm{S}}^\theta(\cdot|x)}\left[r(y)\right]}_{\text{RL objective}} - \alpha\underbrace{\mathbb{E}_{y \sim p_{\mathrm{S}}(\cdot|x)}\big[\mathcal{D}(p_{\mathrm{T}}\|p_{\mathrm{S}}^\theta)(y|x)\big]}_{\text{Generalized On-Policy Distillation}}\Big], \tag{5}$$

where $\alpha \in [0, 1]$ controls the strength of the distillation loss compared to the RL objective. With $\alpha = 1$, it will perform only distillation. The above objective allows us to maximize reward while improving other model capabilities via distillation, which can possibly reduce the "alignment tax" decrease in general model capabilities when aligning language models with human preferences (Ouyang et al., 2022). We apply the above idea to mitigate hallucination using RLAIF, while simultaneously improving downstream performance via distillation (Figure 5).

**Remark**. In RLHF or RLAIF, we typically use reverse KL to constrain the learned policy to stay close to the initial policy. If one wants to only make slight modifications to existing RL fine-tuning workflows, we recommend using reverse KL or JSD $(0.9)$ when integrating GKD with RL.

## 4 EXPERIMENTS

In this section, we evaluate GKD for distilling language models, a typical class of auto-regressive sequence models, on abstractive summarization, machine translation, and arithmetic reasoning.

**Student / Teacher Models**. Our experiments start from student and teacher models with different sizes, specifically open-sourced T5 models (Raffel et al., 2020), that are pretrained on the same datasets. We use supervised fine-tuned T5-XL ($\sim$ 3B params) as the teacher. For students, we use T5-small (77M params), T5-base (250M params), and T5-large (800M params), which are smaller than the teacher by a factor of $38\times$, $12\times$ and $3.8\times$ respectively. See Appendix A.3 for more details.

**GKD Variants**. For choice of divergence $\mathcal{D}$ in GKD in Algorithm 1, we use forward KL, reverse KL and three variants of JSD$(\beta)$: JSD $(0.1)$, JSD $(0.5)$ and JSD $(0.9)$. For student data fraction $\lambda$, we try $\lambda = 1$ (**On-policy**), $\lambda = 0.5$ (**Mixed**) and $\lambda = 0$ (**Supervised**). In particular, we are interested in the on-policy variants ($\lambda = 1$), which have not been previously explored.

**Baselines**. We compare to the widely-used KD methods discussed in Section 3: SeqKD and Supervised KD. We also evaluate ImitKD (Lin et al., 2020) and f-distill (Wen et al., 2023), which can be viewed as "mixed" data variants of GKD ($\lambda = 0.5$) with forward KL and total variation distance as divergence. All the baselines start from the same supervised fine-tuned student checkpoint as GKD.

## 4.1 CASE STUDY: ABSTRACTIVE SUMMARIZATION

We start by evaluating GKD on an abstractive summarization task of generating a summary that captures salient ideas of the input document. To do so, we use the XSum dataset (Narayan et al., 2018), which consists of news articles paired with human-written summaries. Following PaLM (Chowdhery et al., 2022), we evaluate performance using ROUGE-2 score (Lin, 2004) of predicted summaries

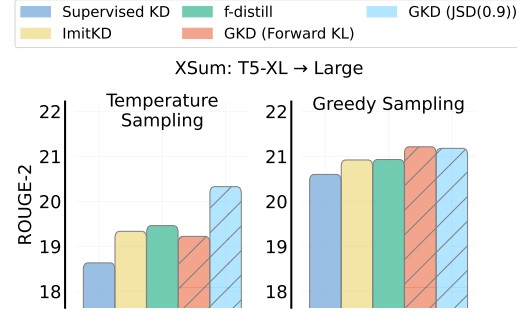

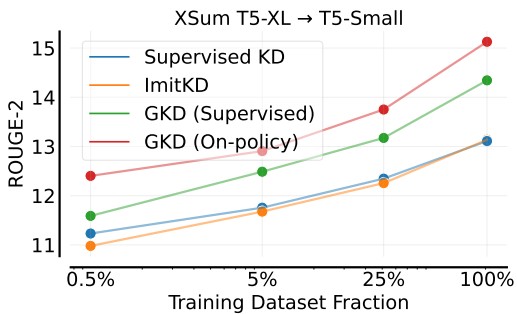

Figure 2: **Comparing GKD to baselines** on distillation from T5-XL to T5-large on XSum. On-policy GKD variants generally outperform baselines.

Figure 3: **Scaling training data**. We evaluate distilled T5-small using temperature sampling ($\gamma = 1$). GKD is more data efficient than baselines.

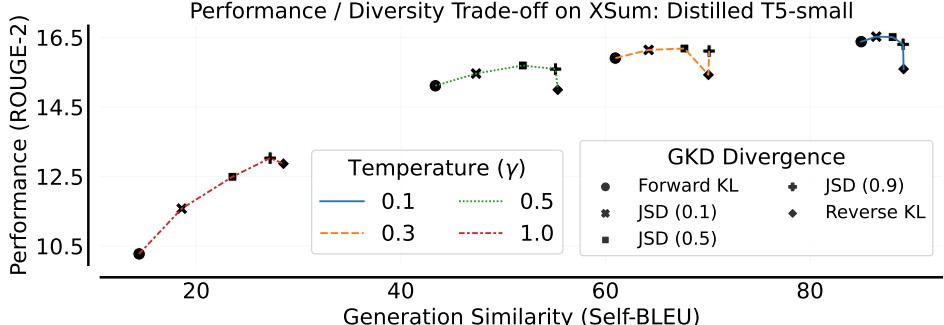

Figure 4: **Effect of Divergence on Performance and Diversity**. Utilizing on-policy GKD with different divergences, we evaluate the trade-off between the distilled student's generation quality and diversity, by varying the sampling temperature. We quantify diversity using Self-BLEU (Zhu et al., 2018), where a score of 100 indicates deterministic outputs and 0 signifies maximum diversity. Transitioning from forward KL to reverse KL, through generalized JSD, leads to decreased diversity, attributed to the enhanced mode-seeking characteristic of the divergence. Mode-seeking divergences often yield superior quality, especially at high temperatures ($\gamma = 1$). Reducing the temperature curtails diversity while narrowing performance differences among divergences.

on the validation split of XSum but observe similar trends in ROUGE-L and ROUGE-1. We use T5 models supervised fine-tuned on XSum as students for distillation while the fine-tuned T5-XL as the teacher. See Appendix A.4 for additional experimental details.

**Comparison to baselines**. First, we explore how GKD compares to widely-used KD approaches, namely SeqKD and Supervised KD, across different student model sizes. As shown in Figure 1, we observe consistent improvements with GKD, which demonstrates the scalability of GKD with respect to the student capacity. Notably, GKD allows us to surpass the few-shot performance of PaLM (540B) using a 7000× smaller T5 model. We also compare GKD variants with ImitKD and f-distill, and evaluate performance with greedy sampling and temperature sampling ($\gamma = 1$) in Figure 2. On-policy GKD with JSD (0.9) outperforms these additional baselines in both scenarios.

**Data efficiency and scaling**. To evaluate the efficiency and scalability of GKD, we distilled the T5-XL teacher using subsampled XSum training datasets: 1K (0.5%), 10K (5%), and 50K (25%) examples. We used T5-small as the student and report data scaling curves in Figure 3. Notably, on-policy GKD on the 5% subsampled dataset, without any ground-truth summaries, outperforms supervised KD and ImitKD with entire training dataset with ground-truth summaries.

**GKD Ablations**. We ablated different divergences and student data fractions in GKD for various student sizes in Figure A.12 and A.13. On-policy and mixed variants consistently outperform supervised variants. Mode-seeking divergences perform better when evaluation is done using temperature sampling while the choice of divergence doesn't affect performance much with greedy sampling.

**Choosing GKD Divergence**. The divergence chosen for distillation is crucial in determining the trade-off between summarization quality and diversity. As the sampling temperature can also be adjusted to balance summary quality and diversity, the optimal choice of divergence is temperature-dependent. To understand this dependence, we evaluate T5-small distilled using on-policy GKD

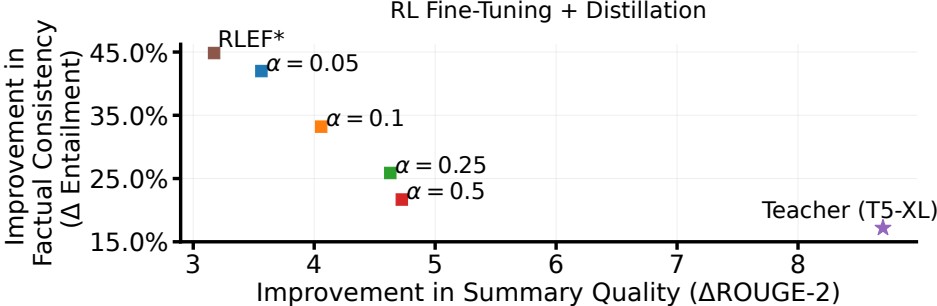

Figure 5: **RLAIF + On-policy GKD**. We show the trade-off between reward maximization and summarization performance on XSum. We report improvements relative to the original T5-base student. Following Roit et al. (2023), we use the textual entailment score from a T5-XXL NLI classifier as the reward. $\alpha$ controls the strength of the on-policy GKD loss with JSD (0.9). As $\alpha$ increases, ROUGE-2 increases while improvement in factual consistency decreases. For comparison, we show the relative performance of the $12\times$ larger T5-XL teacher. RLEF* corresponds to RLAIF method from Roit et al. (2023), where the student is regularized towards the original student model itself instead of the teacher. On-policy GKD + RL achieves higher ROUGE-2 compared to RLEF* while generating more factually consistent summaries compared to the teacher.

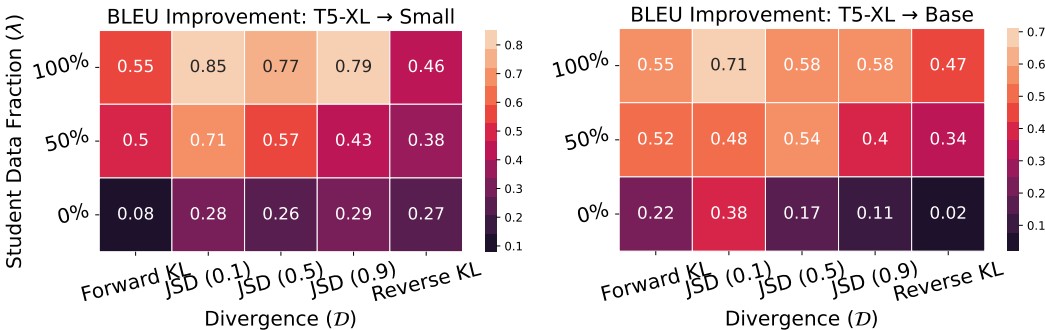

Figure 6: **Varying student data fraction and divergence in GKD on WMT en → de**. For evaluation, we use beam search and report the improvement in BLEU score of distilled student relative to the original student. Results are averaged across three seeds. We observe that using only student-generated output samples outperform other GKD variants. We use the T5-XL (~3B params) supervised fine-tuned on WMT as the teacher, which obtains a BLEU score of 28. **(Left)** We use T5-small (77M params) as the student, which obtain a BLEU score of 25.58. **(Right)** Student corresponds to T5-base (250M params) with a BLEU score of 26.98.

with different divergences. As shown in Figure 4, certain divergences, like JSD (0.5) and JSD (0.9), offer better quality but less diversity at high temperatures. However, as temperature decreases, the difference in quality among divergences narrows, while diversity also drops.

**On-policy GKD with RL**. In summarization, we want model-generated summaries to be factually consistent with their input documents. However, distillation alone might not improve factual consistency as even large models halluncinate and generate inconsistent summaries. Recently, Roit et al. (2023) mitigate hallucination on summarization tasks by using RL with textual entailment feedback as the reward (RLEF), as faithful summaries must be textually entailed from their input documents. Inspired by their success, we explore combining RL fine-tuning using a REINFORCE-like objective with on-policy GKD, as described in Section 3.2. As shown in Figure 5, GKD with RL fine-tuning substantially improves factual consistency compared to the teacher model while obtaining large improvements in summarization quality for the distilled student model.

## 4.2 MACHINE TRANSLATION

To evaluate GKD beyond summarization, we consider the task on translating English to German using WMT14 en-de (Bojar et al., 2014). We report performance on the validation split using the BLEU score, which measures the similarity of machine-translated text to high quality reference translations. We use supervised fine-tuned T5-XL as the teacher with a softmax-temperature of 1.0 (BLEU score of 28). See Appendix A.6 for additional experimental details.

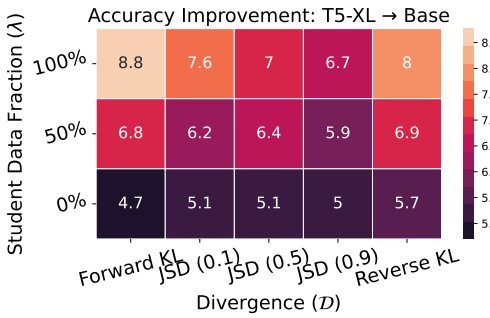
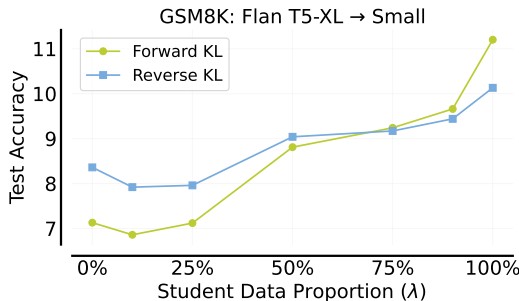

Figure 7: **Ablating GKD on GSM8K.** We distill fine-tuned T5-XL to T5-Base, which obtain an accuracy of 27.9 and 10.16 with greedy sampling.

Figure 8: **Varying on-policy data on GSM8K**. As we increase fraction of student-generated data beyond 25%, performance typically improves.

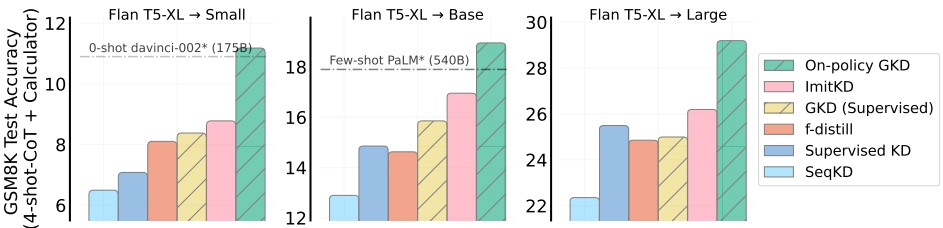

Figure 9: **Distillation on GSM8K with few-shot CoT prompting**. On-policy GKD substantially outperform other approaches. As a reference, we provide GPT-3 davinci-002 results as well as PaLM (540B) results (without a calculator). We use forward KL and reverse KL respectively for on-policy and supervised GKD.

**Results**. Figure 1, A.15 show that on-policy GKD outperforms commonly-used KD approaches. Furthermore, we ablate GKD variants using T5-small and T5-base students in Figure 6. We observe that generalized JSD divergences perform better than forward or reverse KL but their performance gap reduces when using a larger student. Moreover, purely on-policy and mixed data distributions consistently outperform GKD variants only using a fixed supervised dataset, showing the importance of generating sequences from the student. This finding aligns with our results on XSum.

### 4.3 ARITHMETIC REASONING

Wei et al. (2022) show that reasoning abilities only appear to emerge in LLMs with at least several billions parameters, making KD important for improving reasoning abilities of smaller models. To this end, we evaluate GKD on GSM8K (Cobbe et al., 2021), a high-quality dataset of grade school math word problems requiring multi-step logical inference. Here, we explore GKD in conjunction with chain-of-thought (CoT) (Wei et al., 2022), a common approach to improve reasoning abilities of LLMs by prompting them to produce intermediate reasoning steps before giving the final answer.

**Setup**. We perform few-shot prompting by prepending the math problems in GSM8K with the first 4 CoT input-output exemplars from Wei et al. (2022). For evaluation, we report accuracy on the test split by checking whether the target answer matches the final answer given an external calculator, akin to Cobbe et al. (2021). For supervised training, we use the CoT outputs generated by Magister et al. (2022), resulting in around 5.3K (problem, CoTs) pairs in the original training split of GSM8K. We use Flan-T5 models (Chung et al., 2022) supervised fine-tuned for 10K steps on the above CoT dataset as a starting point for distillation. We use the fine-tuned FLAN T5-XL as the teacher, which obtains a test accuracy of 27.9. See additional experimental in Appendix A.5.

**Results**. We first ablate GKD variants and report results in Figure 7 and A.14. We observe that when using only the fixed CoT dataset or mixing it with student-generated CoTs, performance consistently falls short of using solely the student-generated CoTs. Furthermore, forward KL performs quite well, similar to our findings on XSum with greedy sampling. Notably, reverse KL also performs well, especially when training using only a fixed dataset. Additionally, Figure 8 shows that performance consistently improves as the proportion of on-policy data increases, provided that at least 25% of the data is on-policy. Moreover, we demonstrate that on-policy GKD have superior performance compared to baseline KD methods, across all student sizes, as shown in Figure 9.

## 5 RELATED WORK

**Knowledge distillation.** Supervised KD (Buciluǎ et al., 2006; Hinton et al., 2015) is a classic approach and has been successfully used for distilling auto-regressive models (Sanh et al., 2019). Another approach for distilling such models is sequence-level KD (Kim & Rush, 2016). On-policy GKD substantially outperforms supervised KD and SeqKD (Figure 1). Other KD approaches train the student to match different quantities obtained from the teacher, such as hidden states (Jiao et al., 2020) or attention scores (Wang et al., 2020). However, none of these approaches make the connection between distillation and imitation learning, and a purely supervised approach can suffer from train-inference mismatch, also known as exposure bias (Ranzato et al., 2015; Bengio et al., 2015). While He et al. (2019) argue that this mismatch may not be critical, several papers demonstrate that exposure bias leads to poor text generation (Zhang et al., 2019; Chiang & Chen, 2021; Arora et al., 2022).

ImitKD (Lin et al., 2020) identifies this connection by sampling sequences from both the student and a fixed dataset but does not push the idea further. Unlike GKD, ImitKD does not explore purely on-policy data collection, nor does it integrate RL fine-tuning. Moreover, ImitKD keeps the forward KL at the token level, which is not necessary when one has access to the teacher's log-probabilities, rather than just samples. Furthermore, GKD demonstrates the scalability of the idea, handling student models roughly $26\times$ larger than those explored by ImitKD. ImitKD can be viewed as GKD with forward KL and a non-increasing schedule on $\lambda$, a simple choice being $\lambda = 0.5$. More recently, f-distill (Wen et al., 2023) formulates sequence-level KD as minimizing an f-divergence and propose an tractable objective based on total variation distance between the token-level student and teacher distributions. In essence, both ImitKD and f-distill are specific instances of GKD, which we demonstrate lead to worse empirical results than on-policy GKD (Figure 2, 9).

The concurrent work on MiniLLM (Gu et al., 2023) also exploits the link to imitation and frame distillation as an RL problem. In particular, MiniLLM optimizes reverse KL between the teacher and the student at the sequence level (while likelihood maximization is the forward one) using a policy gradient approach. However, we argue that GKD is simpler and more stable, being closer to supervised training, since it does not backpropagate through the student's sampling process. Indeed, MiniLLM relies on a number of stabilizing tricks, to tackle high variance, reward hacking, and generation length bias. GKD is also more general as it can also be used with other divergences such as forward KL or JSD, which can perform better than reverse KL (Figure 6, 7).

**RL fine-tuning.** There are now numerous examples of language models being fine-tuned with RL, be the reward optimizing for some metric (Wu et al., 2018), or learned using human feedback (Ouyang et al., 2022). In these approaches, it is typical to regularize the RL fine-tuned model towards the initial (usually supervised fine-tuned) model. However, as far as we know, we are the first to perform distillation and RL fine-tuning at the same time (Figure 5). If it may seem natural, it is quite different from an optimization perspective, as it changes the regularization towards the initial policy to towards the teacher policy, and we show empirically that it is a viable approach.

**Distillation with reasoning traces or rationales**. Chain-of-Thought prompting (Nye et al., 2021; Wei et al., 2022) has recently demonstrated that LLMs can solve complex reasoning tasks, step by step, just by prompting. This idea was quickly adapted to KD, by extending the teacher dataset with CoT prompts for fine-tuning the student (Magister et al., 2022; Ho et al., 2022; Hsieh et al., 2023). The distillation is still done in a supervised way, and other kind of enhanced prompts could be considered (Li et al., 2022; Mukherjee et al., 2023). We adopt the same approach, but combine it with on-policy distillation with various divergences. It shows the versatility of GKD, and improves upon the purely supervised approaches, as seen in our results on GSM8K (Figure 9).

## 6 CONCLUSION

In this work, we proposed GKD to address the train-inference distribution mismatch and model underspecification when distilling language models. GKD consistently outperformed commonly-used distillation approaches on several generative tasks. We further showed that GKD can be combined with reinforcement learning to optimize a sequence-level reward in addition to distilling the knowledge of a large teacher model, which we believe can improve the widely-used RLHF training phase for language models. One interesting direction would be extending GKD to auto-regressive sequence models for audio (Radford et al., 2023), video (Villegas et al., 2022) and text-to-image generation (Yu et al., 2022). We hope that our work will be valuable for researchers and practitioners who are working on improving performance and efficiency of generative auto-regressive sequence models.

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

# A APPENDIX

## A.1 SELF-DISTILLATION

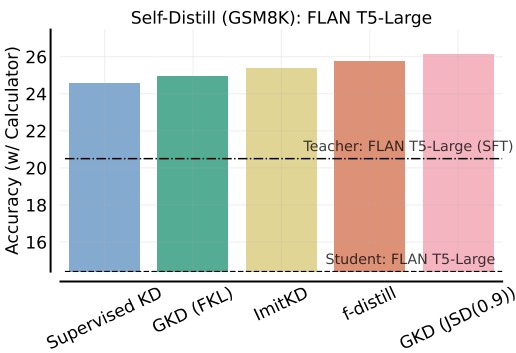

Figure A.10: **Self-Distillation on GSM8K**. Here, GKD corresponds to on-policy GKD ($\lambda = 1$). On-policy GKD variants outperform other approaches including supervised KD. The teacher FLAN T5-Large is supervised fine-tuned on GSM8K and achieves an accuracy of 20.5% while the student FLAN T5-large (not trained on GSM8K) obtains an accuracy of 14.4% on the test set.

Here, we investigate whether GKD works for self-distillation (Yim et al., 2017), where we want to transfer knowledge from a teacher model to a student model with the same architecture and size. To investigate this, we consider self-distillation on GSM8K with FLAN-T5 large as the student and teacher, where the teacher is supervised fine-tuned on GSM8K. As shown in Figure A.10, self-distilled students surpass surpasses the teacher's performance on the test set. Moreover, distillation using student-generated data outperforms supervised KD, with on-policy GKD performing the best.

## A.2 TASK-AGNOSTIC DISTILLATION: INSTRUCTION TUNING

While task-specific distillation provides optimized performance for predefined tasks, which is often crucial for deployment purposes, task-agnostic distillation offers a compelling alternative in scenarios where the exact nature of the task is not known beforehand and can vary during deployment. As highlighted by Sanh et al. (2019), the allure of task-agnostic distillation lies in its efficiency: once distilled, a model can be re-purposed for multiple downstream tasks via prompting or fine-tuning.

**Setup**. To study task-agnostic KD, we focus on instruction tuning (Chung et al., 2022). Our aim is to enhance the distilled model's proficiency to handle diverse tasks presented in the form of instructions. To achieve this, we employ the FLAN T5-XL model as our teacher and distill its knowledge into the FLAN T5-Base, as introduced by Chung et al. (2022). Our distillation process utilizes the comprehensive FLAN2021 instruction tuning dataset, which boasts 5.36 million examples spanning 62 distinct language understanding and generation tasks. For hyperparameter details, see Table A.4.

**Evaluation**. To gauge the versatility of a task-agnostic model, it is essential to test it across a diverse set of tasks. In line with Chung et al. (2022), we evaluate our distilled T5-base student on two held-out benchmark suites: (1) **MMLU** (Massive Multitask Language Understanding) includes exam questions from 57 tasks such as mathematics, history, law, and medicine, and (2) **BBH** (BIG-Bench Hard) includes 23 tasks from BIG-Bench for which PaLM 540B (Chowdhery et al., 2022) performs below average human raters. For performance, we report the distilled model's ability to directly predict the answer via standard few-shot prompting, averaged across tasks in MMLU and BBH.

**Results**. We report the performance of distilled checkpoints obtained after 50K training steps for various methods in Figure A.11. We find that on-policy GKD with reverse KL substantially outperforms supervised KD and ImitKD. Notably, in the context of instruction tuning, we find that using reverse KL performs much better than forward KL. We hypothesize that the efficacy of reverse KL in instruction tuning may stem from its mode-seeking nature as it ensures that the model zeroes in on the main intent or behavior specified by the instruction. As a result, the model might prioritize core behaviors over less relevant details, leading to better performance on held-out tasks.

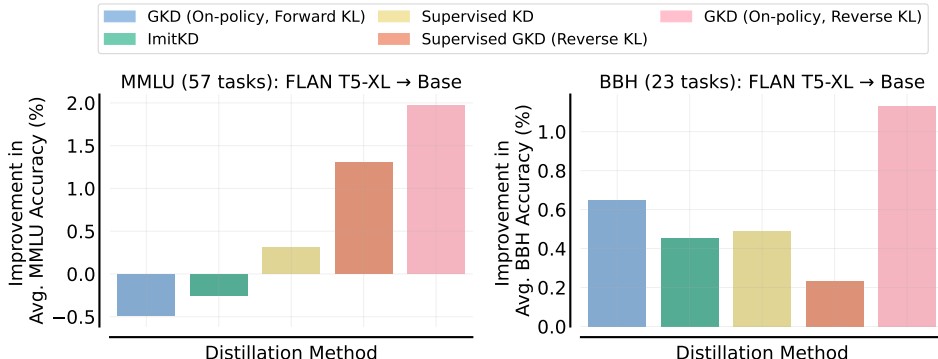

Figure A.11: **Task-agnostic Distillation on FLAN** (Chung et al., 2022). On-policy GKD with reverse KL outperforms other approaches. The evaluation metric on both the MMLU and BBH benchmark suites is few-shot prompted accuracy (exact match), where we take an unweighted average over all tasks. These evaluation benchmarks are held-out (not included in the distillation data). Here, we do not run SeqKD due to its computational inefficiency for generating data from the teacher during training. The teacher FLAN T5-XL achieves an accuracy of 52.4% on MMLU and 41% on BBH, while the student T5-large model obtains an accuracy of 35.6% on MMLU and 31.25% on BBH.

## A.3 T5 MODELS

As base checkpoints, we start from LM-adapted T5v1.1 models. These LM-adapted models are initialized from T5v1.1 and trained for an additional 100K steps on the LM objective discussed in the T5 paper (Raffel et al., 2020). These checkpoints are open-sourced at `https://console.cloud.google.com/storage/browser/t5-data/pretrained_models`.

In our experiments, we initialize both the student and teacher models for distillation by running further supervised FT on the original training dataset, as described below:

- **XSum**. For small, base, large and XL models, we use LM-Adapted T5v1.1 models supervised fine-tuned for 100K, 50K, 38k and 8K steps respectively.

- **WMT**. For small, base, large and XL models, we use LM-Adapted T5v1.1 models supervised fine-tuned for 250K, 250K, 110k and 50K steps respectively.

- **GSM8K**. All models were supervised fine-tuned starting from FLAN-T5 models on the Palm-540 generated CoT dataset for 10K steps.

Similar to T5 and FLAN-T5, our experiments use the Adafactor optimizer (Shazeer & Stern, 2018).

**Computational cost of GKD**. All methods including baselines start from the supervised fine-tuned student checkpoint, which requires training for a few hours on the smallest TPUv3 (8 cores). On GSM8K, the computational overhead from student sampling is approximately $1.8\times$, $2\times$ and $2.2\times$ compared to sampling from a fixed dataset of outputs, for a student-teacher ratio of $38\times$, $12\times$ and $3.8\times$. For RLHF + GKD, the computational overhead is somewhat small as we are only running inference to get teacher logits instead of student logits.

Moreover, the majority of cost in real world use cases is due to serving cost at inference time and not due to fine tuning. Concretely, if it is too costly to sample from the student during fine-tuning, it might also be too expensive to serve this model to users (which may range from tens of thousands to billions). Overall, performance benefits from on-policy GKD might be worth the compute cost, especially when combined with RLHF.

## A.4 XSUM

**Learning rate sweep**. We performed a sweep over the learning rate in {0.0001, 0.0003, 0.001} and found 0.0003 to be the best for T5-base, T5-large and 0.001 leads to best performance for T5-small. As such, by default we use an LR of 0.0003 except when reporting results for T5-small, which uses 0.001. We found that reverse KL was more sensitive to higher LRs and we default to using 0.0003 for all models when using reverse KL.

**Teacher Softmax-temperature**. When using greedy sampling for evaluation, we set the teacher temperature to 1. However, when reporting student performance with temperature sampling ($\gamma = 1$), as done in Figures 2 and 3, we set teacher temperature to 0.1 for the student.

Table A.1: Hyperparameter Details for experiments on XSum.

| Hyperparameter | Value |
|---|---|
| Training Steps | 40,000 |
| Batch size | 32 |
| Eval Split | Validation |
| Dropout | 0.0 |
| Learning Rate (LR) | 0.0003 |
| LR Warmup Steps | 2,000 |
| LR Cooldown (Begin, End) | (30,000, 40,000) |
| Warmup Schedule | Linear (from 0 to LR) |
| Cooldown Schedule | Linear (from LR to 0) |
| Max Input Length | 1024 |
| Max Output Length | 64 |
| Evaluation | Greedy & Temp. Sampling |

**GKD Ablations**. We ablated different divergences and student data fractions in GKD for various student sizes in Figure A.12 and A.13. On-policy and mixed variants consistently outperform supervised variants. Mode-seeking divergences perform better when evaluation is done using temperature sampling while the choice of divergence doesn't affect performance much with greedy sampling.

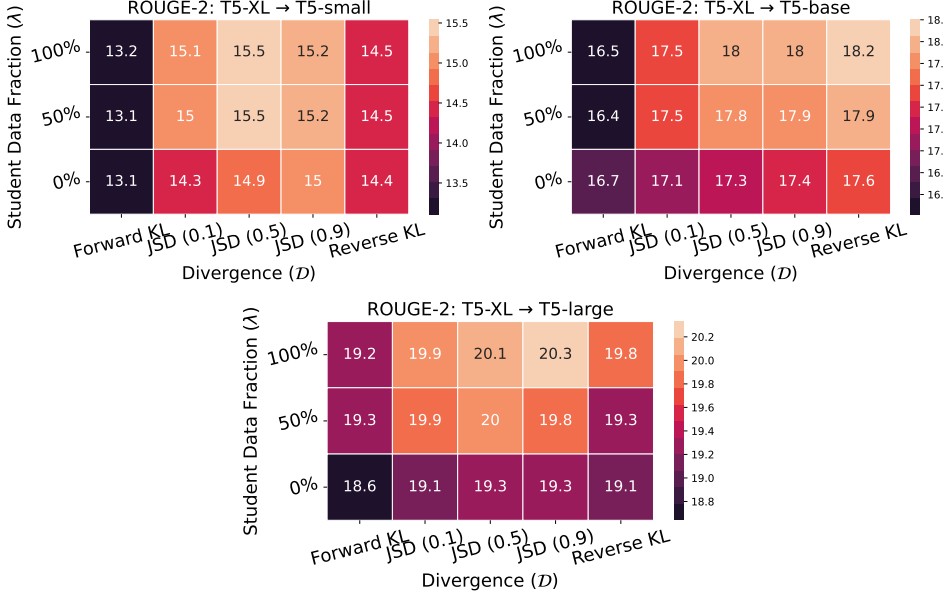

Figure A.12: **Ablating GKD on XSum with evaluation via temperature sampling** ($\gamma = 1$). We distill supervised fine-tuned T5-XL model to different sized student T5 models. Here, we evaluate using temperature sampling and teacher temperature to 0.1 during training. In the plots above, we report the ROUGE-2 score of the student post distillation. On-policy GKD approaches with reverse KL and JSD (0.9) performs the best, while forward KL performs poorly.

## A.5  GSM8K

For training, we use the CoT outputs generated from Palm-540B by Magister et al. (2022). We report accuracy on the original test split of the GSM8K dataset (Cobbe et al., 2021). We use checkpoints at the end of training after distillation for reporting results, which are averaged across 3 seeds.

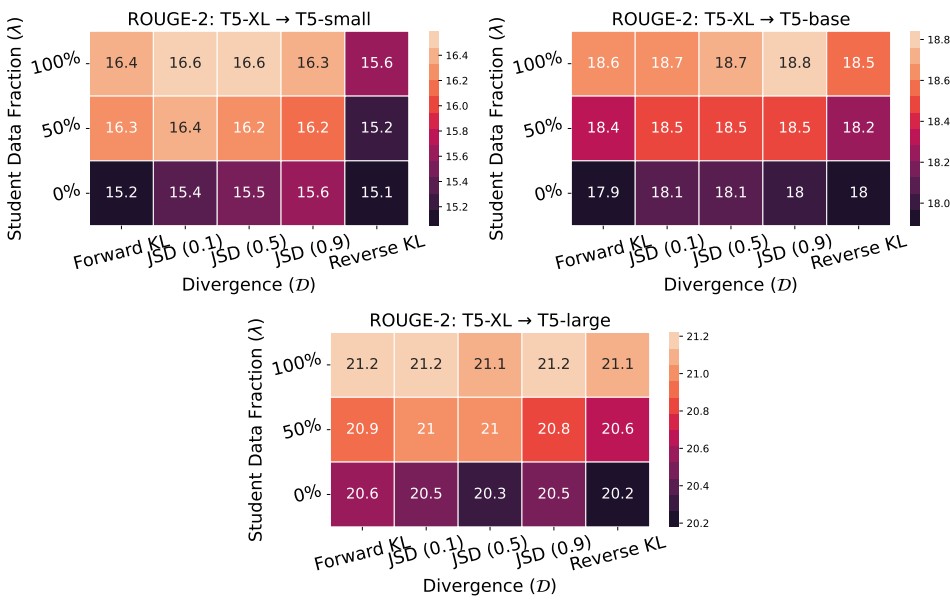

Figure A.13: **Ablating GKD on XSum with evaluation via greedy sampling.** We distill supervised fine-tuned T5-XL model to different sized student T5 models. Here, we evaluate using greedy sampling and set student and teacher temperature to 1 during training. When evaluated using greedy sampling, the teacher model obtains a ROUGE-2 score of 22 while the student T5-small, base and large models obtain score of 13.4, 17.9, and 19.6 respectively. In the plots above, we report the ROUGE-2 score of the student post distillation. On-policy GKD approaches performs the best, with small differences among different divergences. Furthermore, on-policy and mixed variants strongly outperform supervised variants.

Table A.2: Hyperparameter Details for experiments on GSM8K.

| Hyperparameter | Value |
| --- | --- |
| Training Steps | 40,000 |
| Batch size | 32 |
| Evaluation Split | Test |
| Dropout | 0.05 |
| Learning Rate (LR) | 0.0003 |
| LR Warmup Steps | 2,000 |
| LR Cooldown (Begin, End) | (30,000, 40,000) |
| Warmup Schedule | Linear (from 0 to LR) |
| Cooldown Schedule | Linear (from LR to 0) |
| Max Input Length | 512 |
| Max Output Length | 320 |
| Checkpoint | Flan-T5 |
| Teacher softmax temperature | 0.1 |
| Evaluation | Greedy Sampling |

**Few-shot CoT Prompt**. Here, we specify the 4-shot CoT prompt used for the experiments:

Q: There are 15 trees in the grove. Grove workers will plant trees in the grove today. After they are done, there will be 21 trees. How many trees did the grove workers plant today?

A: There are 15 trees originally. Then there were 21 trees after some more were planted. So there must have been 21 - 15 = 6. The answer is 6.

Q: If there are 3 cars in the parking lot and 2 more cars arrive, how many cars are in the parking lot?

A: There are originally 3 cars. 2 more cars arrive. 3 + 2 = 5. The answer is 5.

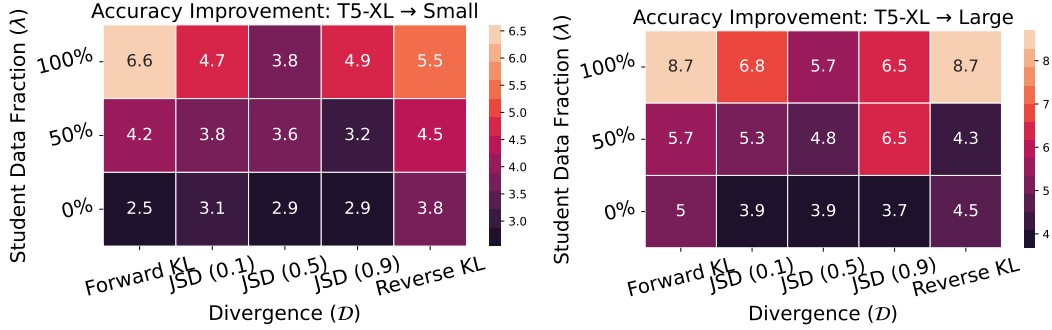

Figure A.14: **Ablating GKD variants on GSM8K with 4-shot CoT**. For evaluation, we use greedy sampling and report improvement in test accuracy of the student after distillation. Results are averaged across three seeds. Using only student-generated output samples typically outperform other GKD variants. We use the supervised fine-tuned T5-XL as the teacher, which obtains an accuracy of 27.9. **(Left)** We use T5-small as the student, which obtains an accuracy of 4.585. **(Right)** Student corresponds to T5-base with an accuracy of 20.5.

Q: Leah had 32 chocolates and her sister had 42. If they ate 35, how many pieces do they have left in total?

A: Originally, Leah had 32 chocolates. Her sister had 42. So in total they had 32 + 42 = 74. After eating 35, they had 74 - 35 = 39. The answer is 39.

Q: Jason had 20 lollipops. He gave Denny some lollipops. Now Jason has 12 lollipops. How many lollipops did Jason give to Denny?

A: Jason started with 20 lollipops. Then he had 12 after giving some to Denny. So he gave Denny 20 - 12 = 8. The answer is 8.

## A.6   WMT

For evaluation, we use beam search with same hyperparameters as Raffel et al. (2020). We report performance of the final checkpoints obtained after training. To reduce the variance in results, we report the results averaged across 3 seeds.

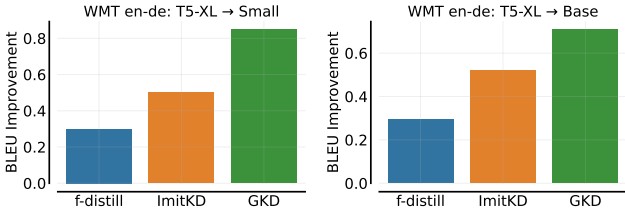

Figure A.15: Comparing GKD to ImitKD and f-distill on WMT. Here, GKD corresponds to best performing variant on WMT, with $\lambda = 1$ (on-policy) and JSD (0.1). On-policy GKD leads to 53% higher BLEU improvement over ImitKD and 162% over f-distill, averaged across small and base models.

Table A.3: Hyperparameter Details for WMT en-de experiments.

| Hyperparameter | Value |
|---|---|
| Training Steps | 100,000 |
| Batch size | 32 |
| Seqio Task Name | 'wmt_t2t_ende_v003' |
| Eval Split | Validation |
| Dropout | 0.0 |
| Warmup Steps | 5,000 |
| Warmup Schedule | Linear (from 0 to LR) |
| Learning Rate (LR) | 0.0003 |
| Input Length (Tokenized) | 80 |
| Output Length (Tokenized) | 80 |
| Teacher softmax temperature | 1.0 |
| Evaluation | Beam Search |

## A.7 INSTRUCTION TUNING

Table A.4: Hyperparameter Details for FLAN Instruction Tuning.

| Hyperparameter | Value |
|---|---|
| Training Steps | 50,000 |
| Batch size | 128 |
| Task | FLAN2021 |
| Dropout | 0.0 |
| Warmup Schedule | No Warmup |
| Learning Rate (LR) | 0.0001 |
| Input Length (Tokenized) | 2048 |
| Output Length (Tokenized) | 256 |
| Teacher softmax temperature | 1.0 |
| Evaluation | Greedy Sampling |

## A.8 MODE-SEEKING VS MODE-COVERING KL

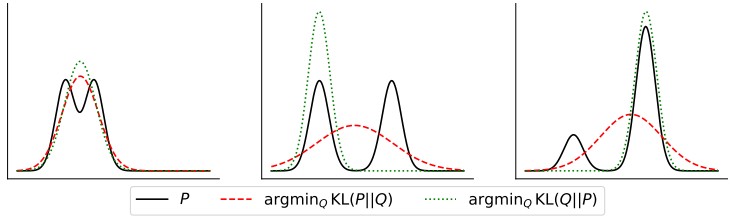

Figure A.16: **Mode-seeking *vs* Model-covering KL with capacity mismatch**. We show the learned distribution $Q_\theta$ when minimizing the forward and reverse KL w.r.t $Q$ between a mixture distribution $P$ and a unimodal Gaussian $Q_\theta$. Reverse KL is mode-seeking as it forces $Q_\theta$ to be zero where $P$ is zero and hence makes it concentrate on one of the modes (last plot). However, forward KL is mode-covering as it ensures that there is some mass under $Q_\theta$ wherever there is some mass under $P$. See Le (2017) to replicate this plot.

