# OpenReview forum: "On-Policy Distillation of Language Models: Learning from Self-Generated Mistakes"
_ICLR.cc/2024/Conference — ICLR 2024 poster_

### Official Review · Reviewer_HokL · 2023-11-01

**Soundness:** 3 good
**Presentation:** 3 good
**Contribution:** 3 good
**Rating:** 8
**Confidence:** 3

**Summary:**

The authors propose generalized knowledge distillation for auto-regressive language models. The authors argue that current form of knowledge distillation does not cater for auto-regressive language models; a student model faces discrepancy between the samples seen in the training and those generated by itself in inference. Inspired by imitation learning, the authors propose on-policy KD, where the student generates a sequence and "imitates" the teacher. In addition, the authors illustrate the proposed on-policy KD can be easily coupled with RL finetuning.

**Strengths:**

- The paper is well-written and easy to follow.
- The proposed distillation method is generalized knowlede distillation that can optimze any divergence between the teacher and student token-level probability distributions.
- The authros illustrate effectiveness of the propose KD on diverse tasks, such as summarization, machine translation, and arithmetric reasoning.
- The proposed method can be easily combined with RLAIF.

**Weaknesses:**

- The authors argue that "auto-regressive sequence models suffer from distribution mismatch between output sequences seen during training and those generated by the student during inference", but this argument requires a thorough citations. There is a contradictory opinion as in [1], where the authors argue that the discrepancy may not be critical.
- The organization can be improved. The authors allocate much portion of this paper on different divergences, yet the use of different divergence seems trivial.
- (minor) in the section 4, "In particular, we are interested in the on-policy variants (λ = 0), which have not been previously explored." should be modified to \lambda=1.

[1] Exposure Bias versus Self-Recovery: Are Distortions Really Incremental for Autoregressive Text Generation?

**Questions:**

- It would be interesting to see if the proposed KD works on self-knowledge distillation, where a teacher and a student are of same size.
- In Appendix Figure A.13, why does the temperature of the teacher bring significant difference in the performance? Temperatures are hardly set to 0.2, but the student works best when temp set to 0.2.

---

> ### Author Response · Authors · 2023-11-16
> **Author response: Added self-distillation on GSM8K, Citations about distribution mismatch, and Clarifications**
>
> We thank the reviewer for taking the time to provide their helpful and valuable feedback. Our response follows:
>
> > **The authors argue that "auto-regressive sequence models suffer from train-inference distribution mismatch", but this argument requires thorough citations. [1] argues that the mismatch may not be critical.**
>
> Thanks for pointing out [1], which we have missed earlier. Since [1], there has been follow-up research which demonstrates the exposure bias leads to poor generation quality and text degeneration [2, 3, 4]. We have added this discussion to the related work.
>
> [2] Bridging the Gap between Training and Inference for Neural Machine Translation. ACL (2019)
>
> [3] Relating neural text degeneration to exposure bias. arXiv preprint (2021).
>
> [4] Why Exposure Bias Matters: An Imitation Learning Perspective of Error Accumulation in Language Generation. ACL Findings (2022).
>
>
> > **It would be interesting to see if the proposed KD works on self-knowledge distillation, where a teacher and a student are of the same size.**
>
> To investigate this, we ran a self-distillation experiment on GSM8K with FLAN-T5 large as the student and teacher, where the teacher is supervised fine-tuned on GSM8K. As shown in Appendix A.2, we find that self-distilled students surpass the teacher's performance on the test set. Moreover, KD approaches using student-generated data outperform supervised KD, with on-policy GKD performing the best with a 5.5% accuracy improvement over teacher.
>
> > **The organization can be improved. The authors allocate much of this paper on different divergences, yet the use of different divergences seems trivial.**
>
> Indeed, different divergences can be easily plugged in GKD, but their impact on performance as well as their task-dependence of the optimal divergence hasn’t been thoroughly explored. That said, based on reviewer's feedback, we moved the figure about mode-seeking vs mode-covering divergences to the appendix. Furthermore, we do plan to include the task-agnostic KD (Appendix A.1)  and self-distillation results (Appendix A.2) to the main paper, but have to figure out the 9 page constraint.
>
>
> > **In Figure A.13, why does the temperature of the teacher bring a significant difference in the performance? Temperatures are hardly set to 0.2, but the student works best when temp is set to 0.2.**
>
> In Figure A.13, we consider a non-standard KD setup where we fix the student temperature to 1 for both training and evaluation on XSum and vary the teacher temperature. Lowering the teacher temperature results in a better performing teacher, which the student tries to mimic by minimizing some divergence to the tempered teacher distribution.
>
> However, in the main paper, we set the student and teacher training temperature to 1 (standard setup) to approximate the true teacher distribution. In this setup, we can vary the student's decoding temperature at inference and find that greedy sampling (temperature = 0) performs much better than using a fixed temperature of 1. We highlight this temperature dependence in Figure 4 where we show the impact of the student's inference temperature on task performance and diversity.
>
> --------
> *We hope that most of the reviewer’s concerns have been addressed and, if so, they would reconsider their assessment. We’d be happy to engage in further discussions.*

---

> > ### Comment · Reviewer_HokL · 2023-11-16
> > **Thank you for the response**
> >
> > You have addressed the concerns in detail, and hence i will update the score.

---

### Official Review · Reviewer_Jqut · 2023-11-01

**Soundness:** 3 good
**Presentation:** 3 good
**Contribution:** 3 good
**Rating:** 6
**Confidence:** 3

**Summary:**

The paper introduces a novel approach called Generalized Knowledge Distillation (GKD) to address distribution mismatch issues in knowledge distillation for auto-regressive sequence models. GKD trains the student model using self-generated output sequences with guidance from the teacher model. It offers the flexibility to employ alternative loss functions between the student and teacher and enables the seamless integration of distillation with reinforcement learning fine-tuning. GKD significantly improves the performance of student models compared to traditional knowledge distillation methods. The paper's contributions include the introduction of GKD, the use of alternative divergence measures, and the combination of distillation with reinforcement learning fine-tuning for the popoular large language models (LLMs).

**Strengths:**

For auto-regressive sequence models, the proposed GKD aims to tackle the distribution mismatch problem between sequences seen during teacher-student training and those generated by the student during inference. GKD, in contrast to traditional methods, trains the student on self-generated sequences with guidance from the teacher.

The paper's strengths are as follows:

1. The paper provides extensive experimental results, demonstrating the effectiveness of GKD in significantly improving student model performance, particularly in tasks like summarization, machine translation, and arithmetic reasoning. The approach is supported by solid empirical evidence;

2. The paper is well-structured, providing clear explanations and insights into the proposed GKD method. The concepts are effectively communicated, making them accessible to the reader;

3. GKD has the potential to enhance knowledge distillation for auto-regressive sequence models, contributing to the development of more efficient and cost-effective models. The paper's findings hold significance in the fields of natural language processing and model compression.

**Weaknesses:**

1. The GKD method has been popular in some related works (Parisotto et al., 2015; Kelly et al., 2019; Agarwal et al., 2022), but it has not been applied to distilled autoregressive models, so the idea of ​​this work is not very novel and original;

2. Although the paper reports significant improvements in performance over baseline KD methods, a more extensive evaluation could strengthen the claims. It would be helpful to see comparisons on a wider range of datasets and tasks to demonstrate the generalizability of GKD;

3. The contributions are not very clear, the authours should explicitly state the key contributions in a dedicated section. This would help readers quickly grasp the paper's core contributions.

**Questions:**

1. In line 10 of Algorithm 1, the gradient calculation of $L_{GKD}$ is incorrect, right? Because that just corresponds to the gradient of a special case of supervised KD when $\lambda$ is 0;

2. I don't quite understand that since the student model itself is smaller and has poorer performance, the quality of the sequences generated by the student model is also not good. So why, judging from the experimental results, is it helpful to introduce output sequences from the student in the training set? Can the authors give a reasonable and detailed explanation?

---

> ### Author Response · Authors · 2023-11-16
> **Author Response: Added results on task-agnostic KD, contributions section, justification for on-policy data, and Clarifications**
>
> We thank the reviewer for taking the time to provide their helpful and valuable feedback. Our response follows:
>
> > **It would be helpful to see comparisons on a wider range of datasets and tasks to demonstrate the generalizability of GKD**
>
> To complement our results on the efficacy of GKD for task-specific distillation on summarization, translation and reasoning tasks, we added a task-agnostic distillation experiment in **Appendix A.1**. Specifically, we distill using the FLAN instruction tuning dataset (5.36M examples) containing several tasks and evaluate it on two held-out benchmarks: MMLU (57 tasks) and Big-Bench Hard (23 tasks). On-policy GKD with reverse KL outperforms common KD approaches (Figure A.10) and achieves +2% and +1% absolute accuracy gain on MMLU and BBH respectively.
>
> > **The authors should explicitly state the contributions in a dedicated section .. to help readers quickly grasp the paper's core contributions.**
>
> We agree and have added the key contributions in the introduction in the revision.
>
> > **GKD has been popular in related works (Parisotto et al., 2015; Kelly et al., 2019; Agarwal et al., 2022), but it has not been applied to distilled autoregressive models, so the idea of ​​this work is not very novel**
>
> While use of on-policy data is popular in robotics and deep RL, our work takes inspiration from these works to introduce GKD for distillation of autoregressive models. Additionally, prior works in RL and robotics mainly focus on forward KL, while we also explore alternative divergences, which could be better in terms of performance and diversity depending on the task. Overall, GKD could be practically significant as it is simple to implement, outperforms prevalent KD approaches, and can be easily combined with RL fine-tuning of autoregressive models.
>
> > **In line 10 of Algorithm 1, the gradient calculation of $L_{GKD}$  is incorrect, right? Because that just corresponds to the gradient of a special case of supervised KD when $\lambda$ is 0**
>
> The gradient is correct – indeed, $\lambda = 0$ corresponds to supervised KD when setting divergence D to forward KL. Different choices of $\lambda$ and divergence lead to other GKD variants. GKD is close to supevised training as it does not backpropagate through the student's sampling process, which makes it quite simple to implement and stable in practice.
>
> > **Why is it helpful to introduce output sequences from the student in the training set during distillation? Provide a detailed explanation**
>
> When training only on a fixed dataset of output sequences, the sequences generated token-by-token during inference by the student can look quite different from the fixed dataset sequences, which we call train-inference distribution mismatch. This mismatch can lead to poor text generation quality [1, 2, 3]. When using student-generated data during distillation, the student receives token-specific feedback from the teacher's logits on the erroneous tokens in its self-generated output sequences. This enables a form of feedback loop akin to what we observe in RL, which helps minimize the train-inference mismatch. Moreover, as the student evolves during training, the quality of data it generates also improves. We have included this in the revision.
>
> For example, on GSM8K, if the student samples a partial solution with some mathematical symbol being incorrect (e.g., using $+$ instead of $\times$), the rest of the solution would also be incorrect. The teacher's feedback can steer the student towards the correct symbol, and in turn towards the correct solution.
>
> [1] Bridging the Gap between Training and Inference for Neural Machine Translation. ACL (2019)
>
> [2] Relating neural text degeneration to exposure bias. arXiv preprint (2021).
>
> [3] Why Exposure Bias Matters: An Imitation Learning Perspective of Error Accumulation in Language Generation. ACL Findings (2022).
>
>
> ----------
> *We hope that most of the reviewer’s concerns have been addressed and, if so, they would reconsider their assessment. We’d be happy to engage in further discussions.*

---

### Official Review · Reviewer_YEFj · 2023-11-03

**Soundness:** 3 good
**Presentation:** 3 good
**Contribution:** 2 fair
**Rating:** 6
**Confidence:** 3

**Summary:**

This paper presents Generalized Knowledge Distillation (GKD), a method that executes knowledge distillation on output sequences generated by both the student and teacher models. The training of the student model utilizes the teacher model's probabilities on the sampled sequences. The proposed method enhances the performance of the student model over standard KD methods across various tasks, including summarization and machine translation.

**Strengths:**

The proposed GKD method is a straightforward and efficient approach for the knowledge distillation of autoregressive models. It demonstrates substantial enhancements across a range of tasks. The paper also introduces a series of ablation studies.

**Weaknesses:**

The paper mainly focuses on task-specific distillation, which requires finetuning the student on the specific tasks first. The two-stage training also brings in additional hyperparameters that need to be adjusted.

**Questions:**

It seems that the GKD method uses the finetuned student as the initialization during task-specific KD. For the other baselines, do you use the same settings?

---

> ### Author Response · Authors · 2023-11-16
> **Author Response: Added results on task-agnostic distillation, Clarifications**
>
> We thank the reviewer for taking the time to provide their helpful and valuable feedback. Our response follows:
>
> > **The paper mainly focuses on task-specific distillation, which requires finetuning the student on the specific tasks first.**
>
> To explore the generality of GKD, we added a task-agnostic KD experiment in **Appendix A.1**. Specifically, we distill using the FLAN instruction tuning dataset (5.36M examples) containing several tasks and evaluate it on two held-out benchmarks: MMLU (57 tasks) and Big-Bench Hard (23 tasks). On-policy GKD with reverse KL outperforms common KD approaches (Figure A.10) and achieves +2% and +1% absolute accuracy gain on MMLU and BBH respectively.
>
> > **The two-stage training also brings in additional hyperparameters that need to be adjusted.**
>
> Indeed, we agree about additional hyperparameters for the SFT phase. That said, the two-stage training is analogous to RLHF, which is widely used for language models, where we first run SFT followed by the online RL fine-tuning phase. As such, two-stage GKD can leverage insights from the two-stage RLHF training as well as be used in conjunction with RLHF with no additional hyperparameters. We have added this in the revision.
>
> > **GKD uses the fine tuned student as the initialization during task-specific KD. For the other baselines, do you use the same settings?**
>
> Yes, all baselines start from the SFT student checkpoint. We mention this explicitly in the revision.
>
> -----
> *We hope that most of the reviewer’s concerns have been addressed and, if so, they would reconsider their assessment. We’d be happy to engage in further discussions.*

---

### Official Review · Reviewer_Qwi5 · 2023-11-03

**Soundness:** 2 fair
**Presentation:** 3 good
**Contribution:** 2 fair
**Rating:** 6
**Confidence:** 3

**Summary:**

This paper proposes GKD for auto-regressive LM distillation, aiming to address the distribution mismatch between the output sequence during training and inference. GKD trains the student on its self-generated output sequences. GKD significantly improves student’s performance over commonly used KD methods for autoregressive LMs.

**Strengths:**

1. The paper proposes a unified framework for existing KD methods within the context of on-policy imitation learning.
2. GKD can be easily combined with RLHF. This allows the student to learn capabilities from the teacher while aligning with humans.
3. The authors investigate different divergence objectives and student data fractions on various tasks, providing practical insights.

**Weaknesses:**

1. GKD has limited technical novelty upon ImitKD/f-distill. ImitKD/f-distill adopts similar imitation learning ideas to GKD, except that GKD makes $\lambda$ tunable and explores different divergences. The latter is more of a hyper-parameter choice as it highly depends on target tasks and decoding strategies.
2. To verify the initial claim that GKD alleviates the train-test distribution mismatch issue, the authors may consider include an experiment with a mismatched test domain.
3. Missing comparison between GDK and ImitKD/f-distill in NMT.
4. GKD seems to be computationally expensive due to the need of fine-tuning a good student initialization and sampling from student distribution. What is the computational overhead?

**Questions:**

See weakness.

**Details Of Ethics Concerns:**

No ethics concerns.

---

> ### Author Response · Authors · 2023-11-16
> **Author Response: Added Task-Agnostic KD with held-out benchmarks, Comparisons on NMT, and Clarifications**
>
> We thank the reviewer for taking the time to provide their helpful and valuable feedback. Below are responses to their concerns:
>
> > **Missing comparison between GKD and ImitKD/f-distill in NMT.**
>
> We have added this comparison in the **Appendix A.6**. On-policy GKD leads to 53% higher BLEU improvement over ImitKD and 162% over f-distill, averaged across small and base models.
>
> > **To verify .. that GKD alleviates train-test distribution mismatch, the authors may consider including an experiment with a mismatched test domain.**
>
> The train-test distribution mismatch (also known as exposure bias in NLP) corresponds to mismatch between “fixed” output sequences seen during model training vs sequences generated auto-regressively by the model at inference, irrespective of the test domain. To clarify this in the revision, we renamed it to “train-inference” distribution mismatch.
>
> Nevertheless, to explore the generality of GKD, we added a task-agnostic distillation experiment in **Appendix A.1**. Specifically, we distill using the FLAN instruction tuning dataset (5.36M examples) containing several tasks and evaluate it on two held-out benchmarks: MMLU (57 tasks) and Big-Bench Hard (23 tasks). On-policy GKD with reverse KL outperforms common KD approaches (Figure A.10) and achieves +2% and +1% absolute accuracy gain on MMLU and BBH respectively.
>
> >  **ImitKD/f-distill adopts similar imitation learning ideas to GKD, except that GKD makes $\lambda$ tunable and explores different divergences.**
>
> Indeed, both ImitKD and f-distill can be viewed as special cases of GKD. However, on-policy GKD ($\lambda=1$), which hasn’t been explored prior to our work, outperforms both ImitKD and f-distill across several tasks. Additionally, the combination of GKD with RLHF is novel. Furthermore, our work highlights the task-dependent nature of optimal divergence as well as the importance of on-policy data, which could be useful for practitioners.
>
> > **GKD seems to be computationally expensive due to the need of fine-tuning a good student initialization and sampling from student distribution. What is the computational overhead?**
>
> All methods including baselines start from the supervised fine-tuned student checkpoint, which requires training for a few hours on the smallest TPUv3 (8 cores). The computational overhead from student sampling is approximately $1.8\times$, $2\times$ and $2.2\times$ compared to sampling from a fixed dataset of outputs, for a student-teacher ratio of $38\times$, $12\times$ and $3.8\times$. For RLHF + GKD, the computational overhead is somewhat small as we are only running inference to get teacher logits instead of student logits.
>
> We believe that the majority of cost in real world use cases is due to serving cost at inference time and not due to fine tuning. Concretely, if it is too costly to sample from the student during fine-tuning, it might also be too expensive to serve this model to users, which may range from tens of thousands to billions. Overall, performance benefits from on-policy GKD might be worth the compute cost, especially when combined with RLHF. We have added this discussion in Appendix A.3.
>
> -----
> *We hope that most of the reviewer’s concerns have been addressed and, if so, they would reconsider their assessment. We’d be happy to engage in further discussions.*

---

> > ### Comment · Reviewer_Qwi5 · 2023-11-22
> >
> > Thanks for the authors for providing task-agnostic distillation experiments and clarifications on their contributions with respect to previous works. I have no remaining concerns.

---

### Author Response · Authors · 2023-11-20
**Common response and updates for revision**

We thank the reviewers for their valuable feedback! All reviewers recommend acceptance and found the paper to be well-written, empirically strong with extensive experiments and ablations, and of significance to knowledge distillation and NLP community. Based on reviewers' comments, we made the following changes:

- Generalizability of GKD beyond task-specific distillation **[Reviewer Qwi5, YEFj, Jqut]**
    - We added a task-agnostic distillation experiment in **Appendix A.1**. Specifically, we distill T5-XL to T5-small using the FLAN instruction tuning dataset (5.36M examples) containing several tasks and evaluate it on two held-out benchmarks: MMLU (57 tasks) and Big-Bench Hard (23 tasks). On-policy GKD with reverse KL outperforms common KD approaches (Figure A.10) and achieves +2% and +1% absolute accuracy gain with direct few-shot prompting on MMLU and BBH respectively.

- Added GKD results for self-distillation on GSM8K and citations showing train-inference mismatch leads to poor generation **[Reviewer HokL]**

- Discussion about compute cost of student sampling and similarity of two-stage RLHF training to GKD **[Reviewer Qwi5]**

- Added key contributions in introduction and improved explanation for on-policy KD **[Reviewer Jqut]**

---

### Public Comment · ~Jiaye_Wu1 · 2026-02-25
**Question regarding potential notation error in eq 4**

Eq 4 seem to have redundant integration. There is already an integration over $y$ inside $\mathcal{D_{KL}}(p_T \| p^\theta_s)(y|x)$. Therefore, is the $\mathbb{E}_{y \sim p_s(\cdot | x)}$ redundant as it is integrating over $y$ again?

---

### Meta-Review · Area_Chair_LbY6 · 2023-12-10

**Metareview:**

This paper presents Generalized Knowledge Distillation (GKD). Contrary to sequence-level KD (Kim & Rush 2016) where the labels are generated according to the teacher's sequence distribution, or supervised KD (Hinton et al. 2015) where the labels are taken from ground truth, here the algorithm samples with a some probability the labels from the student's (autoregressive) sequence distribution. The training of the student model utilizes the teacher model's probabilities on the sampled sequences.

This shows strong results in summarization, NMT, grade school maths for distilling a 3B params encoder-decoder autoregressive LM in 77M, 250M, 800M models. The limitations of the work are mostly in breadth of the experimental validation and in the fact that previous works in non-autoregressive models basically follow the same method. The authors made a thorough job of answering the reviewers' questions.

Overall, the paper is suitable for ICLR and I recommend acceptance with a poster.

**Justification For Why Not Higher Score:**

The novelty is not particularly high (see https://openreview.net/forum?id=3zKtaqxLhW&noteId=dSXampbRhz & https://openreview.net/forum?id=3zKtaqxLhW&noteId=BQKfAdOPdB ), the method lacks a few experimental ablations/validations (https://openreview.net/forum?id=3zKtaqxLhW&noteId=dSXampbRhz).

**Justification For Why Not Lower Score:**

This works well already, better than all other methods of KD in their experiments. Particularly relevant as it's for autoregressive models in a world of LLM.

---

### Decision · Program_Chairs · 2024-01-16

Accept (poster)